# A Modified U-Net Model for Predicting the Sea Surface Salinity over the Western Pacific Ocean

Xuewei Zhang [1], Ning Zhao [2] and Zhen Han [1,3,*]

1 College of Marine Sciences, Shanghai Ocean University, Shanghai 201306, China; d190200040@st.shou.edu.cn
2 Japan Agency for Marine-Earth Science and Technology, Yokosuka 2370061, Japan; zhaoning@jamstec.go.jp
3 Shanghai Engineering Research Center of Estuarine and Oceanographic Mapping, Shanghai 201306, China
* Correspondence: zhhan@shou.edu.cn

**Abstract:** The prediction of oceanic features is always an important issue in oceanography, where deep learning has been proven to be a useful tool. In this study, we applied the improved U-net model to predict the monthly sea surface salinity (SSS) over the western Pacific (WP) Ocean, and the model was designed to use the SSSs from six consecutive months to predict the SSS in the next month. The monthly satellite-based SSSs in 2015–2020 were used for model training, and the data collected after January 2021 were used to evaluate the model's predictive abilities. The results showed that the predicted SSSs represented the general patterns of SSSs over the WP region. However, the small-scale features were smoothed out in the model, and the temporal variations were also not well captured, especially over the East China Sea and Yellow Sea (ECS&YS) region. To further evaluate the potential of the U-net model, a more specific model was conducted for the ECS&YS region (Domain 2), which successfully predicted both spatial and temporal variations in the SSSs, including the spreading and retreating of the low-salinity tongue. Based on the comparison between the two domains and sensitivity experiments, we found that the prediction biases were contributed by the spatial distributions of the SSSs, the domain size, and the filter numbers. In addition, further multi-step prediction experiments suggested that our U-net model could also be used for long-time prediction, and we have examined up to five months. Overall, this study demonstrated the great ability and potential of the U-net model for predicting SSS, even though only a few trainable data are available.

**Keywords:** deep learning; ConvLSTM; U-net; sea surface salinity; western Pacific; prediction





## 1. Introduction

In recent decades, with the development of both sensors and algorithms, observations of large-scale oceanic features have been promoted to a high level of accuracy and coverage [1,2]. More recently, large-scale sea surface salinity (SSS) values also became available via satellite measurements, which greatly improved our understanding of oceanic processes, such as the tropical instability waves [3], large-scale intra-seasonal to interannual variabilities [4–7], the Earth's water cycle [8], and marine ecosystems [9]. Moreover, considering the importance of SSS in environmental monitoring, recent studies have also focused on obtaining more accurate and real-time predictions with computational procedures [10–12]. Specifically, compared with statistical (or empirical) methods [12], the recent 'hot spots' have been in machine learning-based approaches, which produce benefits from their advantages in representing the nonlinear and stochastic phenomena [11,13].

Deep learning, as a specific machine learning method, was developed on the basis of surface learning by imitating the information processing neural structure of the human brain to extract features from external input data, thus enabling a machine to understand the learning data and obtain specific information [14]. Particularly, as one of the deep learning models, the recurrent neural networks (RNNs) were designed to extract dynamic

time series and temporal features through the context of events (e.g., [15]), which have great potential in predictions. To fix the gradient vanishing/exploding problems (i.e., the gradients in the RNN may grow/decay exponentially to infinity or zero due to the long-term dependencies) in practical applications [16–18]. Hochreiter et al. (1997) revised the traditional RNN model and proposed the long short-term memory (LSTM) network in the late 1990s [19]. In recent decades, the LSTM networks have been widely used in many fields, including the predictions of short- and mid-term oceanic features [20–22].

Moreover, to extract features that are varying both spatially and temporally, Shi et al. (2015) introduced the convolutional LSTM (ConvLSTM) [23], which combined the original LSTM and the convolution networks that were designed for handling images [24]. Recent studies have further demonstrated that the ConvLSTM could not only consider the spatiotemporally varying features but also predict their future changes [25–27], which makes the ConvLSTM a powerful tool for oceanic studies [28,29]. For example, Xu et al. (2020) suggested that the ConvLSTM has higher accuracy and generalization capability than the traditional prediction models [30], and other studies have also shown that the ConvLSTM could be used in three-dimensional oceanic fields [31,32].

Although deep learning models exhibit great advantages in predicting oceanic features, they also have some issues to be overcome, such as the lack of interpretability compared to the regular physical models, the lack of clear routines for model optimization, the loss of extreme values, and most importantly, the requirement of large amounts of data for model training, e.g., [13,21,33]. Most studies on predictions using deep learning models have focused on the sea surface temperature (SST) based on long-term spot observations or satellite observations that became available in the 1990s [11,34]. Compared to these—due to the relatively late advances in retrieving SSSs [35,36]—far fewer datasets are available today; for this reason, few attempts have been made to predict SSSs using deep learning models.

Recently, a newly proposed architecture of convolutional networks, the U-net model, has demonstrated an outperformed ability to analyze spatially varying patterns [37]; this model has been well-tested in studies on clouds or oceanic eddies [38–42]. By replacing the ordinary convolutional filters in the U-net model with the ConvLSTM filters, the U-net model has also become a powerful tool for predictions [43]. More importantly, compared to the original ConvLSTM models, the U-net model may require fewer trainable data to achieve the precious information [44], and recent studies have also shown that the U-net model has better prediction performance in terms of accuracy, better recognition continuity, and richer fault detail [45]. These key advantage of the U-net model is essential for SSS predictions based on satellite observations, which motivated this study.

Thus, the goal of this study was to evaluate the potential ability of the U-net model to predict the spatiotemporal variations in SSSs based on a few available trainable data. In addition, instead of showing the well-tuned results only, this study also aimed to provide detailed procedures for error reductions, which would help the readers optimize the U-net or other similar deep learning models for their own applications. In this study, we conducted a set of experiments for SSS predictions based on the U-net model, with most of the ordinary convolution filters replaced by the ConvLSTM filters [43,44]. The study domain was chosen to cover the western Pacific Ocean (1°S–45°N, 121–160°E; see Section 2.3), which contains most of the common oceanic features—such as western boundary currents [46], oceanic fronts [47,48], and river-dominated marginal seas with significant spatiotemporal variations in salinity [49,50]—thus providing a great stage for evaluating the performance of our U-net model.

## 2. Materials and Methods

### 2.1. Data Source

In this study, the monthly SSS data were obtained from the Soil Moisture Active Passive (SMAP) L3 salinity dataset from April 2015 to March 2022 with a spatial resolution of 0.25° [51]. The SMAP satellite [35,52,53] is an orbiting satellite, launched by the National Aeronautics and Space Administration (NASA), that can simultaneously monitor soil

moisture and SSS globally; it carries an L-band microwave radiometer and adopts a multi-polarization mode with a non-sky bottom incident angle of 40° and a revisit time of 8 days. The SMAP L3 SSS products are created by averaging all the valid L2C observations within each grid using the 8-day running mean, and the monthly data were subsequently obtained [54]. In our experiments, the data from 2015 to 2020 were employed for training the U-net model, and the data collected after January 2021 were used to test the predictability of our model.

### 2.2. The U-Net Model

Following the general concept of the U-net model [55,56], our U-net model for SSS prediction contains two phases: the encoding phase and the decoding phase (Figure 1). The encoding phase is a series of continuous down-sampling processes used to extract feature information. This phase consists of three steps with different filters in each step, and each filter contains two 3 × 3 convolution layers (followed by the ReLU activation function) and a 2 × 2 maximum pooling layer. The decoding phase contains a similar structure but with up-sampling processes, while the last convolution layer is a 1 × 1 convolution operation followed by a sigmoid activation function. Specifically, in our U-net mode, most of the convolution layers that were used in the original U-net model were replaced by the ConvLSTM layers to obtain the skills necessary to learn the temporal signals.

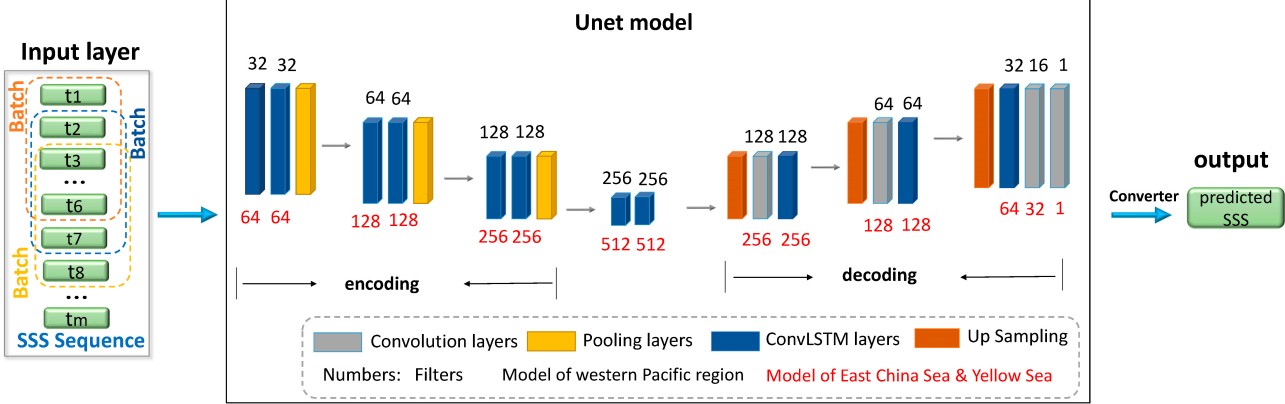

**Figure 1.** Structures of the modified U-net model and illustrative study flow.

The ConvLSTM layer enables the extraction of both spatial and temporal features during the convolutional computation. In addition, the ConvLSTM layer uses gate structures to extract and store useful information [57]. Particularly, the convolution kernels, $W_i$, $W_f$, and $W_o$, slide through the window of the two-dimensional spatial matrix, and the convolution result may vary from 'input', 'updates the cell state', 'forgotten', and 'output' according to the gate structure. The formulas of the ConvLSTM are as follows.

$$i_t = \sigma(W_{wi} * x_t + W_{hi} * h_{t-1} + W_{ci} \circ c_{t-1} + b_i) \tag{1}$$

$$f_t = \sigma\left(W_{xf} * x_t + W_{hf} * h_{t-1} + W_{cf} \circ c_{t-1} + b_f\right) \tag{2}$$

$$c_t = f_t \circ c_{t-1} + i_t \circ tanh(W_{xc} * x_t + W_{hc} * h_{t-1} + b_c) \tag{3}$$

$$o_t = \sigma(W_{xo} * x_t + W_{ho} * h_{t-1} + W_{co} \circ c_t + b_o) \tag{4}$$

$$h_t = o_t \circ \tanh(c_t) \tag{5}$$

where $i_t$ indicates input gates; $f_t$ indicates forget gates; $c_t$ indicates the cell state; $o_t$ indicates output gates; $h_t$ is the hidden layer at time $t$; $W$ is the weight matrix; $b$ is the offset from the input gate to the output gate; and $\circ$, $\sigma$, $*$, and tanh denote the Hadamard product operator, the logistic sigmoid operator, the convolution operator, and the hyperbolic tangent operator, respectively.

Overall, the procedures of the U-net model can be summarized as follows. During the encoding phase, the size of the image is continuously compressed, but the number of feature channels increases, and the extracted features are more abstract, richer, more expressive of the target, and more robust in performance. On the other hand, during the decoding phase, the original image accuracy is restored through deconvolution to recover detailed information.

*2.3. Model and Domain Settings*

In this study, we focused on the SSSs over the western Pacific (WP) region (121°–160°E, 1°S–45°N; Figure 2). Both open oceans and marginal seas are included in this region, and the SSSs vary in very different ways within these regions. For example, as shown in Figure 2A, the mean SSSs over the central Pacific are generally higher than 34 psu, but the seasonal variations are quite small. On the other hand, the SSSs become relatively lower in marginal seas, especially the East China Sea and Yellow Sea (ECS&YS) region, which has a mean SSS of only about 30 psu. In particular, the low salinity region over the ECS&YS region was dominated by the Changjiang diluted water, where the actual SSSs could vary from less than 6 to over 40 psu throughout the year, thus leading the pronounced seasonal cycle [58,59] (Figure 2B).

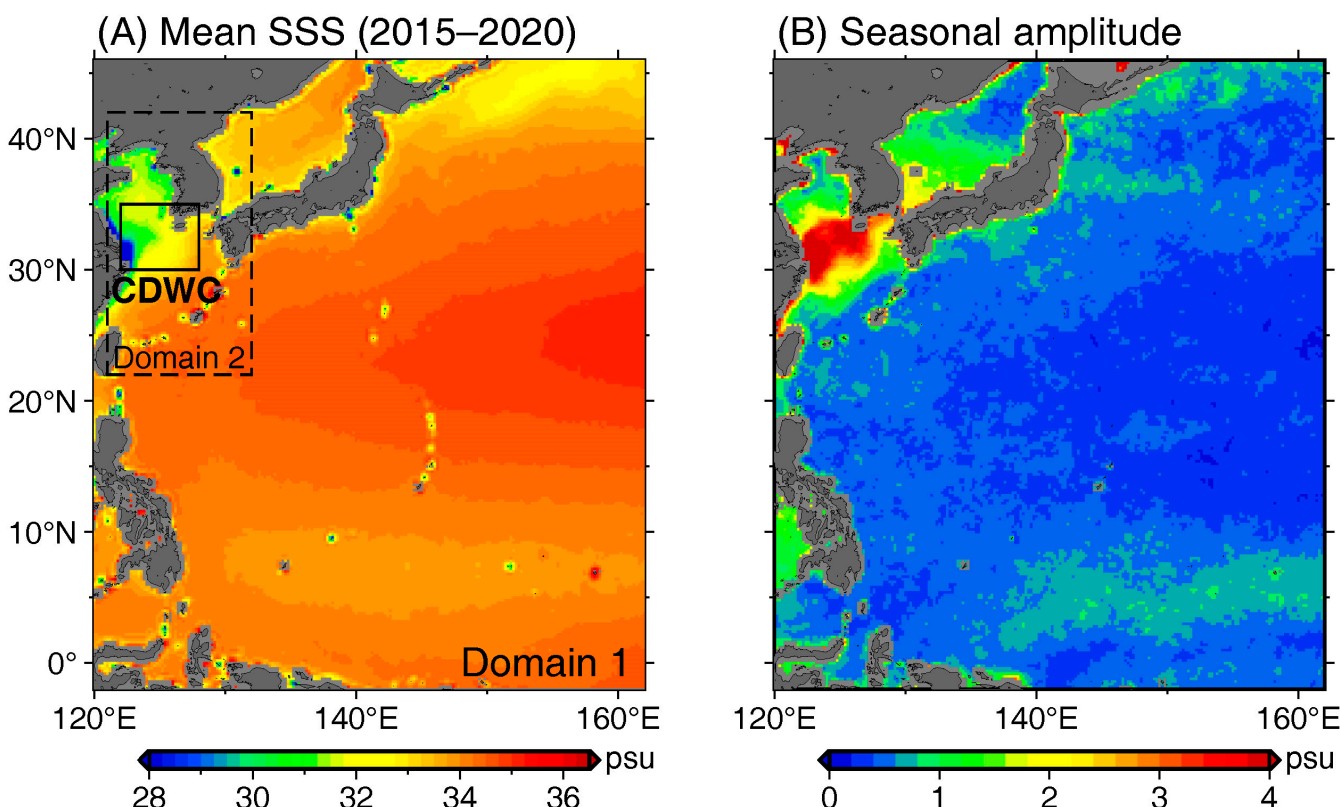

**Figure 2.** (**A**) Mean SSS from 2015 to 2020 and (**B**) its seasonal amplitude over Domain 1 based on satellite observations; the region of Domain 2 is marked as the dashed box. The seasonal amplitude was obtained by averaging the differences between the maximum and minimum of the SSSs in each year from 2015 to 2020. The Changjiang Diluted Water and its Continuums (CDWC; 122~128°E, 30~35°N) is also marked by the black box in panel (**A**).

Two models were constructed in this study, which covered the WP (i.e., Domain 1) and the ECS&YS (i.e., Domain 2) regions, respectively. We used 32 filters in the first layer of the U-net model for the WP region and 64 filters for the ECS&YS region. The filters in the following layers were then defined accordingly (Figure 1). To find the optimal batch size, we performed four experiments using five, six, seven, and eight consecutive months

as input and found the six-month batch size was the best (see Figure S1). Moreover, the choice of filter number is discussed in Section 4.2.

After the filter number and batch size were determined, we trained the model three times to ensure the model was stable and robust (see Supplementary Table S1 for the three trials), and the first model for each domain was used for the following analyses. In addition, we also tested two commonly used models, the LSTM and ConvLSTM, to ensure that the U-net model did have the ability to predict SSSs (see Figure S2 for comparisons).

### 2.4. Pre- and Post-Processing

To reduce the potential influences of the spatial characteristics of different regions, we used the SSS anomalies—which were obtained by subtracting the long-term mean SSSs of 2015–2020—to train the models instead of the original SSSs. After that, all SSS anomalies were further rescaled into 0–1 using the min–max normalization method to prevent distribution jumps during training and to accelerate the convergence rate:

$$x_{norm} = \frac{x - Min}{Max - Min} \tag{6}$$

where $x$ is the gridded monthly SSS anomaly, $x_{norm}$ is the normalized SSS anomaly, and *Max* and *Min* are the maximum and minimum SSS anomalies in 2015–2020, respectively.

Accordingly, the model output could be restored to the normal SSS by reversing the normalization and adding back the long-term mean. In particular, considering the nature of the LSTM filter (that contains the 'forget' gate), the model tended to fit the majority of the training data and led to the underestimation of extreme values [33]. For example, our model was inclined to ignore the changes in the CDWC region, which only covers a small area but has much lower salinity and stronger variations than the open oceans. As a result, after denormalization, the predicted maximum and minimum SSS anomalies were smaller than the values of Max and Min used in Equation (6). Therefore, we further tuned the model output with an extra 'denormalization' by stretching the predicted SSS anomalies to match with the range of the true SSS anomalies in 2015–2020 before adding the mean values back.

Note that we extrapolated the SMAP SSS over the land area before the training stage by solving Poisson's equation via an iterative relaxation scheme because our U-net model could not be trained with the missing values. The SSSs over land were then removed from the output data based on the original land–sea masks extracted from the input data.

### 2.5. Model Validation

To evaluate the model performance, we used the mean error (ME), the mean absolute percentage error (MAPE), the root-mean-square error (RMSE), and the pattern correlation coefficient (PCC), which can be obtained as follows [33,60].

$$ME = \overline{y_{pred}[i] - y_{true}[i]} \tag{7}$$

$$RMSE = \sqrt{\frac{\sum_{i=1}^{N}\left(y_{pred}[i] - y_{true}[i]\right)^2}{N}} \tag{8}$$

$$MAPE = \left(\sum_{i=1}^{N}\left|\frac{y_{pred}[i] - y_{true}[i]}{y_{true}[i]}\right|\right)/N \times 100 \tag{9}$$

$$PCC = \frac{\sum_{i=1}^{N}\left(y_{true}[i] - mean\left(y_{true}[i]\right)\right)\left(y_{pred}[i] - mean\left(y_{pred}[i]\right)\right)}{\sqrt{\sum_{i=1}^{N}\left(y_{true}[i] - mean(y_{true}[i])\right)^2}\sqrt{\sum_{i=1}^{N}\left(y_{pred}[i] - mean\left(y_{pred}[i]\right)\right)^2}} \tag{10}$$

where $y_{true}[i]$ is the true SSS at the *i*th grid, $y_{pred}[i]$ is the predicted SSS, the overbar in Equation (7) represents the time mean, and $N$ is the total grid number.

## 3. Results

### 3.1. The Western Pacific

Figure 3 represents the spatial distributions of the true and predicted SSSs during the four seasons of 2021. It was found that, in most regions, the SSSs were highest in the spring and became lower from summer to autumn. Compared with the true values, our U-net model captured the general features of SSSs, such as the westward extension and eastward retreating of the salty water from the central Pacific and the meridional low–high–low SSS patterns. Meanwhile, as shown by the error indices, our model performed almost uniformly over both the open ocean and marginal seas, where the mean errors were lower than ±0.5 psu in most regions. The RMSEs and the MAPEs were about 0.7 psu and 1.0%, respectively, and showed little change across different seasons.

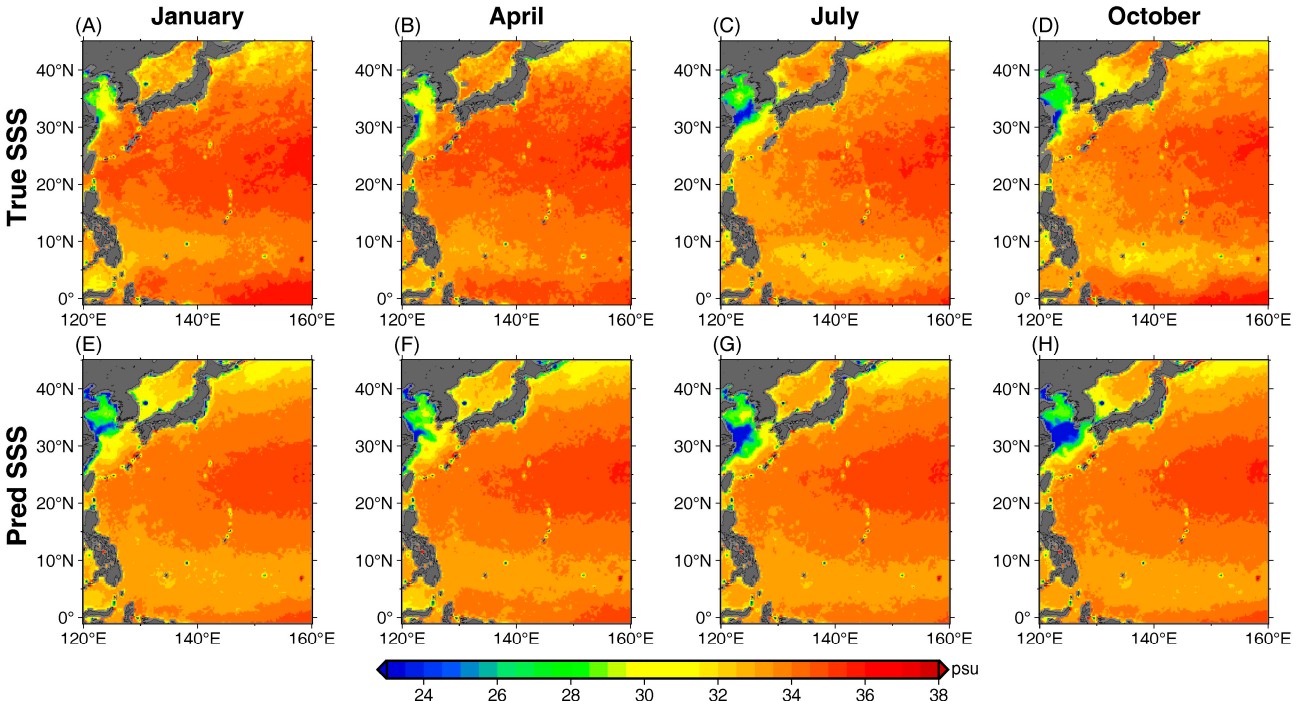

**Figure 3.** Spatial distributions of true (**A–D**) and predicted SSSs (**E–H**) for 2021 over the western Pacific region.

Compared to the area-averaged indices, the PCCs had a larger seasonal amplitude, while the values were only around 0.6~0.75. These relatively lower correlations suggest that the Domain 1 model did not predict the spatial distributions of SSSs well enough, which smoothed out some small-scale features. For example, the zonal-orientated low SSS zone over the northern equatorial counter-current region (NECC, 5~8°N) and the high-salinity region south of it were not well represented, resulting in even worse performance in representing the temporal changes in those regions.

Specifically, large errors were also found in the ECS&YS region (Figure 4A), where negative biases covered almost the entire ECS, especially over the so-called 'Changjiang diluted water and its continuums (CDWC)' region [61] (Figure 2), and some signals even spread eastward to the Tsushima Strait. By comparing the temporal variations in the predicted and true SSSs, we found that the Domain 1 model had almost a constant negative bias for most months, and the largest errors appeared in autumn when the model failed to predict the re-salting processes from July [62]. Meanwhile, although the model did predict some signals of surface freshening over the CDWC region, it overestimated the freshening and had a lag of one month due to the barely learned temporal variations. As a result, the RMSEs and MAPEs over the CDWC region became almost three times larger than

those averaged over Domain 1 (Figures 4B and 5B), especially during cold seasons (autumn and winter).

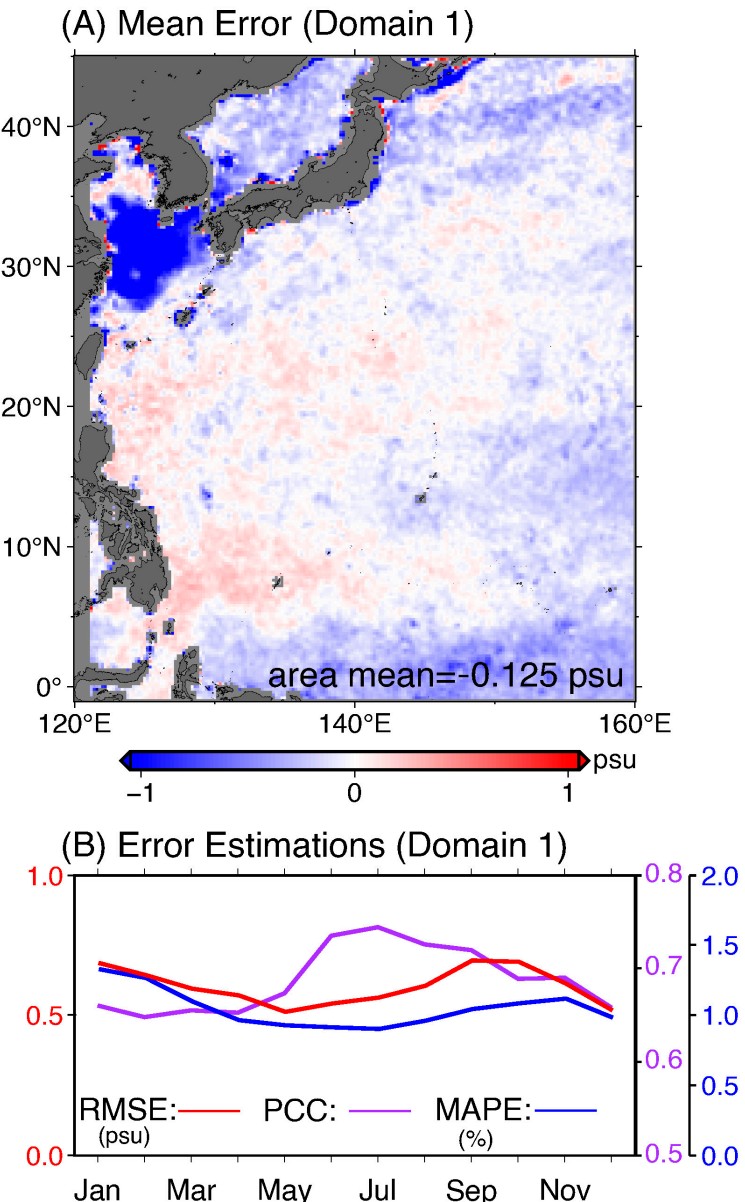

**Figure 4.** (**A**) Annual mean error of the predicted SSSs in 2021 over Domain 1 and (**B**) the monthly variations of the error estimations: root-mean-square error (RMSE), the pattern correlation coefficient (PCC), and mean absolute percentage error (MAPE).

Overall, the results of the Domain 1 model suggest that the U-net model does have some potential in predicting the SSSs; however, it might be difficult for it to capture both the general pattern and the detailed variations over a large domain with limited trainable data. Therefore, to further evaluate whether the U-net model could be used in regions with both large spatial and temporal variations, we constructed another U-net model for the ECS&YS region (i.e., Domain 2 in Figure 2), where the model showed the worst performance.

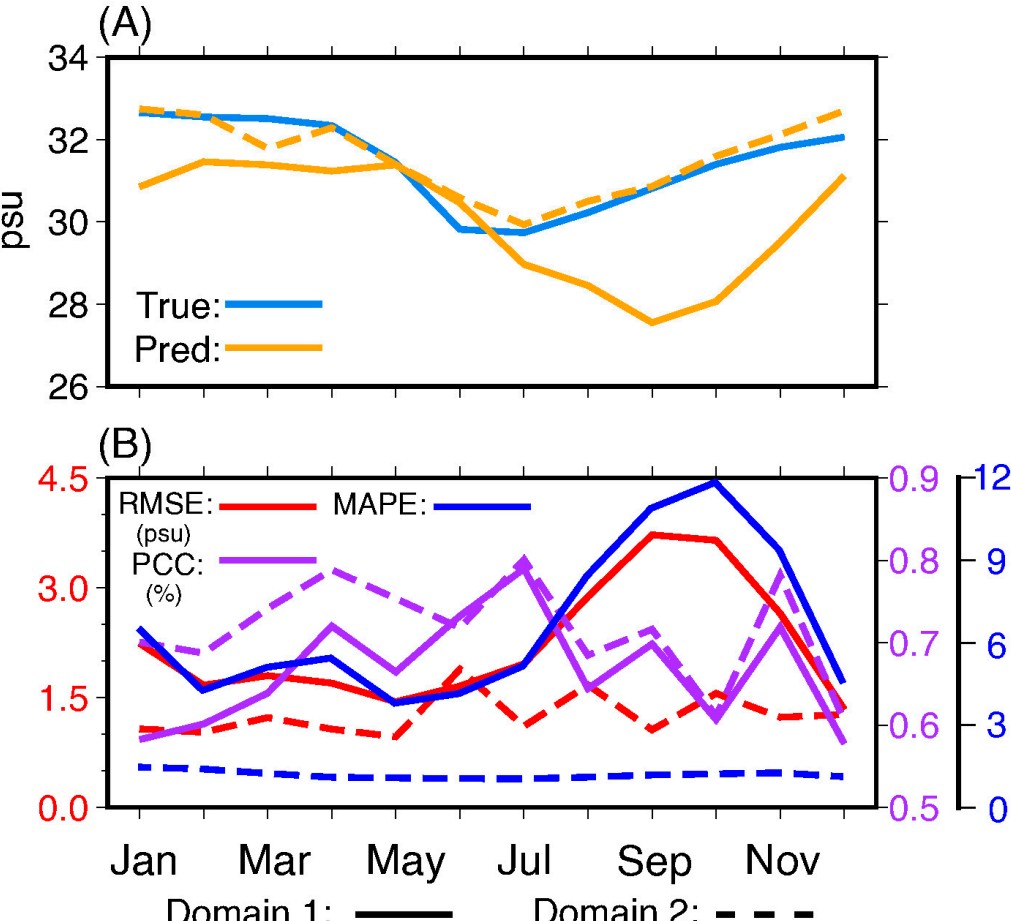

**Figure 5.** Temporal variations in true and predicted SSSs (**A**) together with the error estimations (**B**) over the CDWC region.

### 3.2. The East China Sea and the Yellow Sea

Before checking the detailed results of the Domain 2 model, our first concern was the seasonal variations in the predicted SSSs, which were barely predicted in Domain 1. As shown by the dashed lines in Figure 5, the Domain 2 model showed good agreement with the true SSSs, as both the freshening and the re-salting were captured well. Because of that, RMSEs and MAPEs were largely reduced, especially for the cold seasons. Meanwhile, the PCCs also slightly increased in the spring. Figure 6 shows the total error estimations over Domain 2. Instead of the large negative errors, our new Domain 2 model had positive biases in most regions, and the total mean error was only about 0.3 psu. Moreover, compared to the results from Domain 1, the area-averaged RMSEs and MAPEs were also reduced by 30% (from 1.7 psu and 3.6% to 1.1 psu and 2.5%, respectively), while the PCC also increased from 0.67 to 0.71 (Figure 6B,C). Note that the performance of the Domain 2 model is comparable to or even better than the previous studies at predicting SSS in coastal regions [43,63], and the prediction errors are also close to the SMAP SSS dataset itself, whose RMSEs were about 0.5~4 psu in coastal regions (within 300 km away from the coastline) [64].

After confirming the overall performance, our next target was the spatial distributions of the predicted SSSs during the freshening and re-salting periods (from May to October). As shown in Figure 7A–F, as the volume transport of the Yangtze River increases in the summertime rainy season, the low-SSS tongue (i.e., the CDWC) from the river mouth spread over a large area that covers the whole YS and the northern ECS. Some of the low-salinity water passed through the Tsushima Strait and entered the Sea of Japan (Figure 7C–E; also see Figure 3C,D) before the low SSS tongue started to retreat in August. After September, the low salinity water was trapped again near the coastal regions (Figure 7E). Compared

to the true variations, the Domain 2 model well predicted the evolutions of the CDWC with realistic spatial distributions and temporal evolutions (Figure 7G–L), including the low-salinity water spreading from the river mouth from May and its retreat in August.

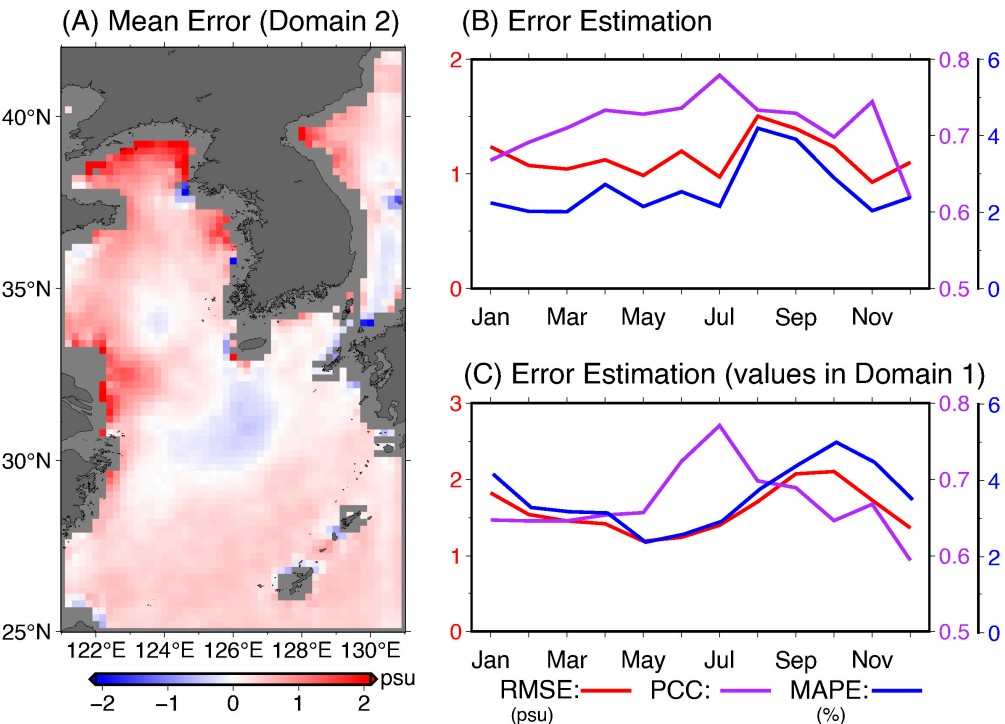

**Figure 6.** (**A**) Annual mean error of the predicted SSSs in 2021 over Domain 2 and (**B**) the monthly variations of the error estimations. For ease of comparison, error estimations using data from Domain 1 are also plotted in panel (**C**).

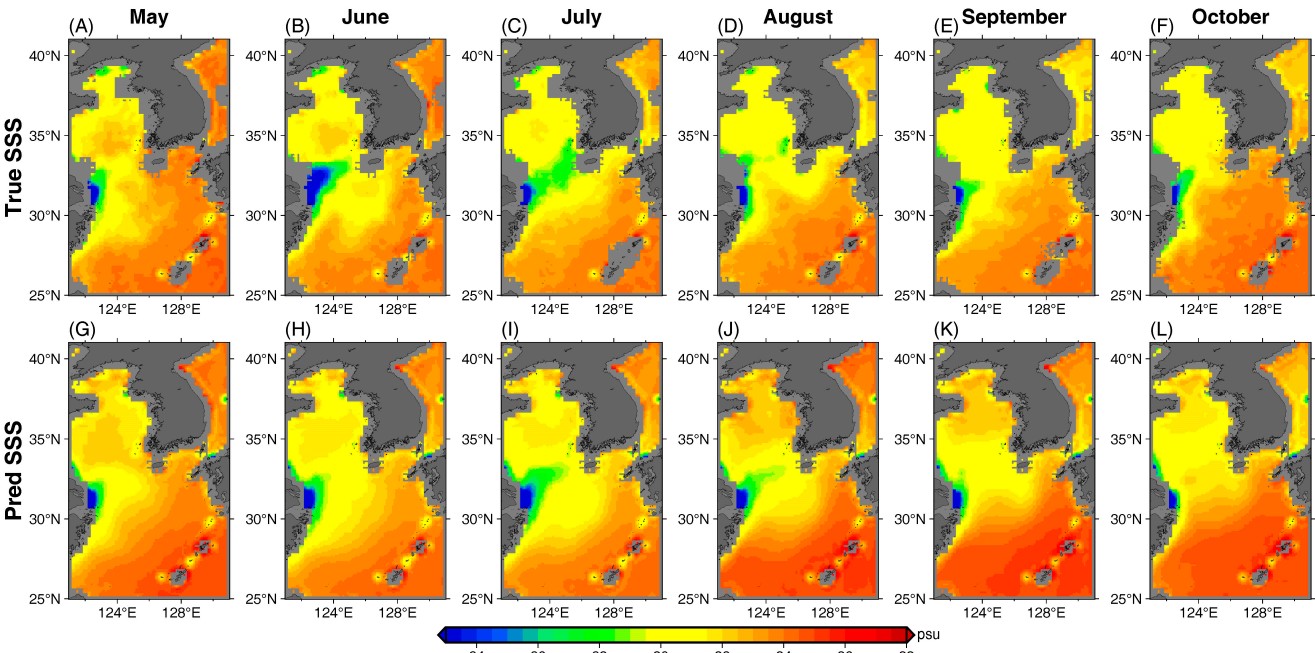

**Figure 7.** Spatial distributions of true (**A–F**) and predicted (**G–L**) SSSs (Domain 2) for the summer of 2021 over the East China Sea and the Yellow Sea region.

On the other hand, the Domain 2 model did have some relatively larger errors during August and September (Figure 6), as shown by the unrealistic salty conditions near the coastlines over the Bohai Sea and the southwestern side of Domain 2 around 122°E, 33°N. Such coastal biases were likely induced by both the performance of the satellite-based SSSs (e.g., lower spatiotemporal resolution and lesser accuracy, [64]) and the extrapolated 'pseudo-SSS' over the land during the training stage. The simplest way to reduce these is to use datasets with higher spatial and temporal resolutions (see Section S1 in Supplementary Materials for additional experiments on the impact of the temporal resolution of the training data); however, it requires further experiments and better datasets for model training, which will be the focus of one of our future works.

In summary, the above results confirmed that the U-net model containing replaced ConvLSTM filters and proper settings does have the ability to learn and predict oceanic features with large spatial and temporal variations. Note that we also confirmed the above conclusion by conducting an extra experiment for the badly predicted NECC region mentioned in Section 3.1, and the new experiment showed the U-net did predict the spatial and temporal variations over that region (see Figures S6 and S7).

## 4. Discussion

### 4.1. Optimization of Model Performance Using Training Data

Considering the nature of the ConvLSTM filter, which is designed to retrieve the common patterns and 'forget' the rare samples, it is reasonable to consider that the large spatial differences among the SSSs over the WP region may reduce our model performance (e.g., Figure 2A). Following that, in this study, we used the SSS anomalies from the long-term mean in the model training to reduce the potential influences. However, it remains unclear whether such influences exist and how large they could be. Therefore, to evaluate this, we performed another experiment using the same model settings for the two domains but using the original SMAP SSS data.

Figure 8 shows the standard errors of the predicted SSSs in the two domains using different training data. In general, our current model shows better performance than the models trained using original SSSs. The removal of the long-term mean SSSs in the training data reduced prediction errors by about 0.1~0.2 psu over the NECC region and the Kuroshio and its extension region, while some larger error reductions could also be seen for coastal areas in the ECS&YS regions.

Interestingly, over the region dominated by the CDWC, our pre-processing approach seemed to make the results even worse. Such degradation was likely due to the Domain 1 model not capturing the CDWC patterns at all, as suggested by the fundamental prediction errors after June (Figure 5A; also see Figure S3B). In contrast, in Domain 2, the pre-processing approach did help the model correctly 'learn' the behavior of the CDWC, which significantly reduced prediction errors (over 3 psu) in this area and the adjacent Tsushima Strait. On the other hand, we noticed that the pre-processing approach also caused slightly larger errors south of the river mouth (around 123°E, 28°N in Figure 8F) in Domain 2, where the Taiwan Warm Current dominated [65]. Thus, the errors may be caused by our limited domain size, which did not include the Taiwan Strait and, therefore, could not capture the upstream structures of the Taiwan Warm Current.

Note that the use of SSS anomalies may cause some overall biases (e.g., Figure S3A) because the mean SSS field also varies across the years (i.e., the interannual variations). One possible solution is to build another model for predicting the interannual variations in SSSs; however, it is beyond the scope of the current study, and it is difficult to build with the limited data available in this study (only six years from 2015 to 2020).

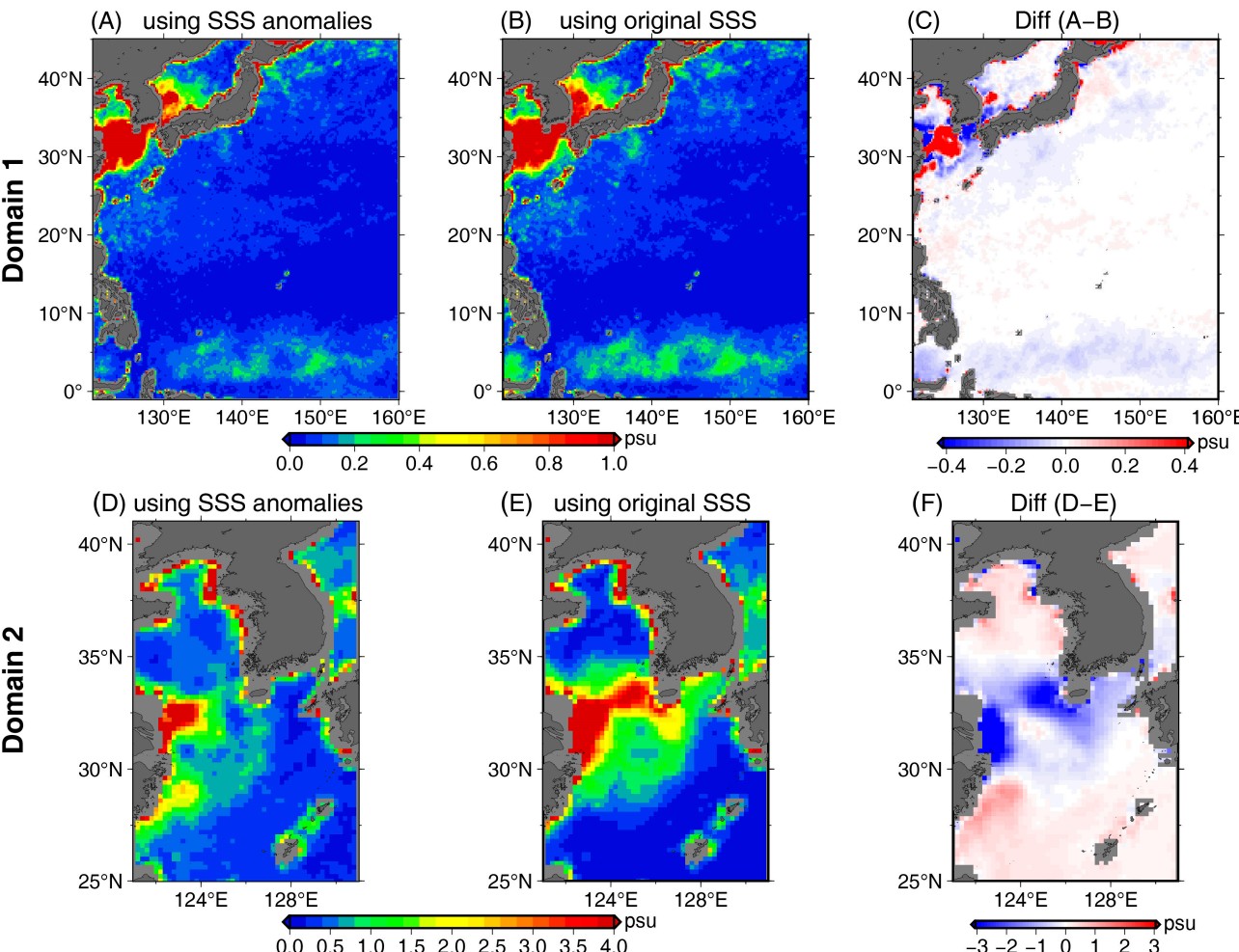

**Figure 8.** Standard errors of predicted SSSs in two domains based on models using (**A**,**D**) SSS anomalies and (**B**,**E**) original SSSs. The differences between the two standard errors are shown in panels (**C**,**F**), and the negative values show improvement. Note that the upper panels for Domain 1 use a smaller range in color shading for representing the improvement over the open seas.

*4.2. Influences of the Filter Numbers*

As we mentioned in Section 2.3, we used 32 filters in the first layer in the Domain 1 model; however, it failed in predicting the SSS variations over the ECS&YS region. In contrast, with the incorporation of the doubled filter numbers, the Domain 2 model successfully predicted the SSSs with greatly improved reproducibility of the CDWC. Thus, one may consider that such an improvement may mainly be caused by the larger filter numbers we used in Domain 2 compared to Domain 1.

To investigate this, we adopted a set of sensitivity experiments for both Domain 1 and Domain 2 with different filter numbers in the first layer. As shown in Figure 9A, all Domain 1 models failed to catch the seasonal cycle of the SSS over the CDWC region. Moreover, predictions became even worse when we used 64 filters in Domain 1, suggesting that more filters would not have led to better performance, especially in predicting relatively small features within a large domain. Meanwhile, by evaluating the performance of the Domain 2 model with 32 and 128 filters, we further confirmed that, even in the smaller Domain 2, more complex structures would not lead to better results (Figure 9B; also see Figure S4 for the values averaged over the entirety of Domain 2).

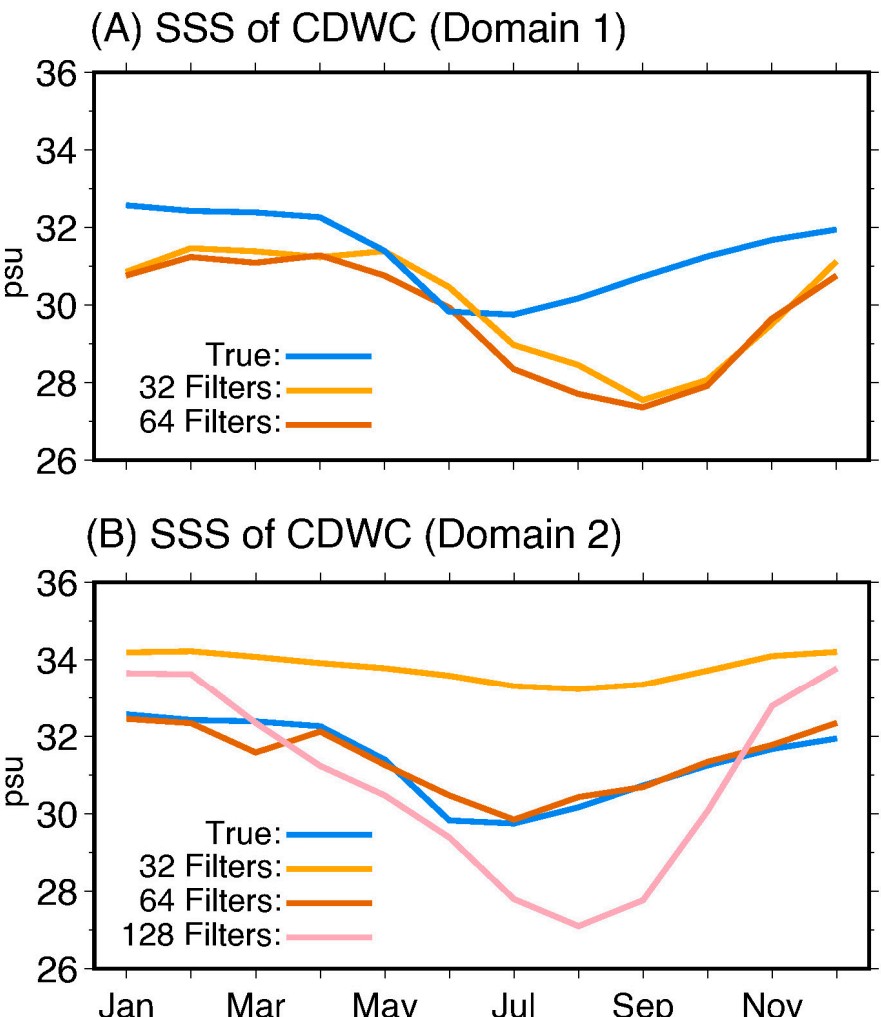

**Figure 9.** Comparison of the true and predicted SSSs in the CDWC region using the (**A**) Domain 1 and (**B**) Domain 2 models with different filters.

Consequently, our results suggest that there would be optimal settings for the U-net model for specific domains or target features, and the domain size may be more important in predicting features with strong temporal variations and relatively small spatial scales [66] than the filter numbers of the U-net model. Similar conclusions could be obtained for other regions (see Figures S7 and S8 for an extra experiment in the NECC region). Note that we only conducted the comparison with filter numbers, and more experiments may be needed to quantitatively evaluate the potential influences of the domain size or the area ratios between the target feature and the domain size.

*4.3. Model Performance in Multi-Step Prediction*

Unlike the previous studies on the SSTs or other quantities (e.g., [29–31,67]), the data availability limited the training settings. In this study, the model was originally designed to predict 1 month into the future. However, theoretically, the model could predict infinite time series of SSSs if we replace the input time series with the previously predicted SSSs. Therefore, of particular interest to us is examining how far our model could predict using the current training dataset and settings.

Figure 10 shows the error estimations of the predicted SSSs for the 1-month (i.e., the current settings), 2-month, 3-month, 4-month, and 5-month predictions, respectively. Interestingly, the multi-step predictions were quite stable and acceptable, although errors became slightly larger after the two-step predictions. Moreover, the results showed that the

performance of the Domain 1 model did not change much in multi-step predictions, and the increase in errors was almost neglectable. Such performance was likely caused by the small seasonal amplitudes in most regions of Domain 1 (Figure 2B); however, this was not the case over the ECS&YS region. As shown by the pink lines in Figure 10, the averaged errors increased by about 15% from the two-step predictions over the ECS&YS region, and they remained at similar levels in the following steps. Comparatively, the Domain 2 model showed even larger errors in multi-step predictions, which increased by over 40% from the second step. However, it should be noted that the prediction in Domain 2 was still better than that in Domain 1, as shown by the smaller RMSEs and higher PCCs.

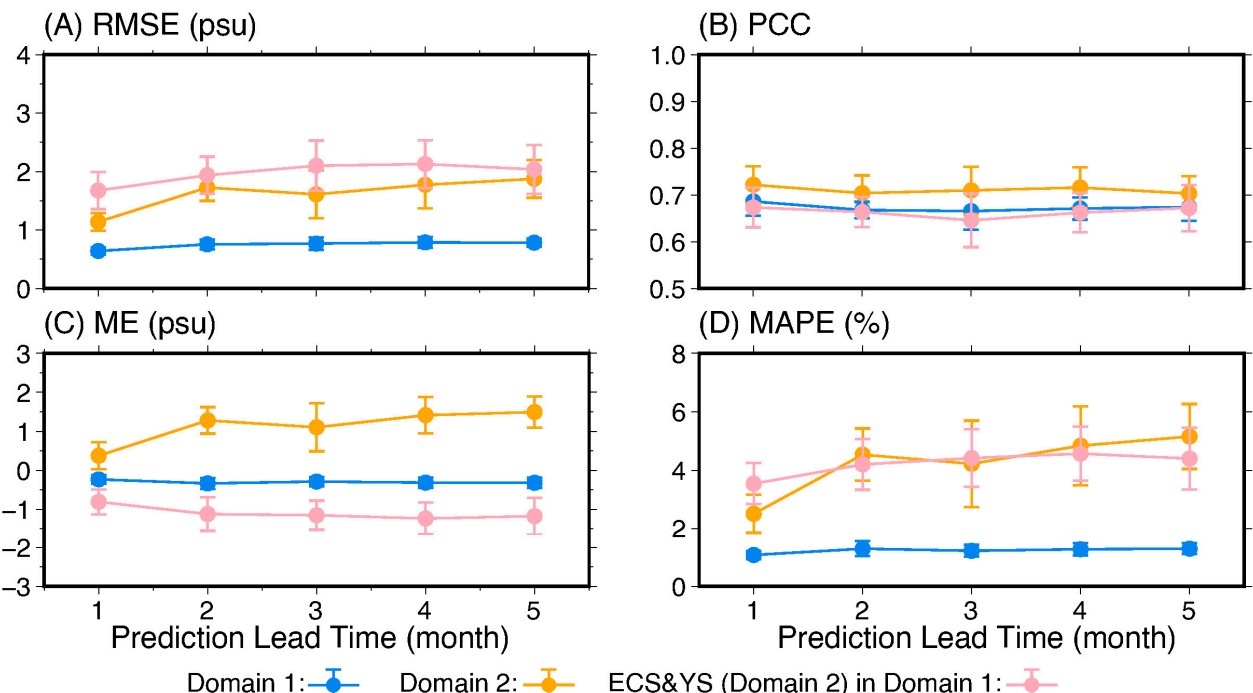

**Figure 10.** Error estimations of multi-step prediction over Domain 1 (blue) and Domain 2 (orange), and the results for the ECS&YS region in Domain 1 (pink) are also plotted for comparison. Colored dots represent the mean error indices of all the predicted months, while the error bars represent their standard deviations. Note that, due to data availability, the one-month-lead prediction contains 15 months (ended in March 2022), and the five-month-lead prediction only contains 11 months.

To determine why the prediction errors in the CDWC largely increased, we plotted the SSSs predicted using the two-step prediction (Figure 11). Compared to the one-step prediction (Figure 7), the low-salinity tongue that expanded from the river mouth of the Yangtze River almost disappeared, inducing a large underestimation of the SSSs over the CDWC region. It was found that the current model, trained by the monthly dataset, could not predict the surface freshwater intrusion without the precursor at least one month before. Such a conclusion could be further confirmed by our extra experiments based on the semi-monthly data (obtained through the linear interpolation of the monthly data). Usually, linearly interpolated data should not be used for model training. However, our results suggest that such procedures could be useful in multi-step predictions with monthly datasets because the interpolated semi-monthly data provided 'pseudo' intermediate conditions between two consecutive months (see Section S1 and Figures S5 and S6).

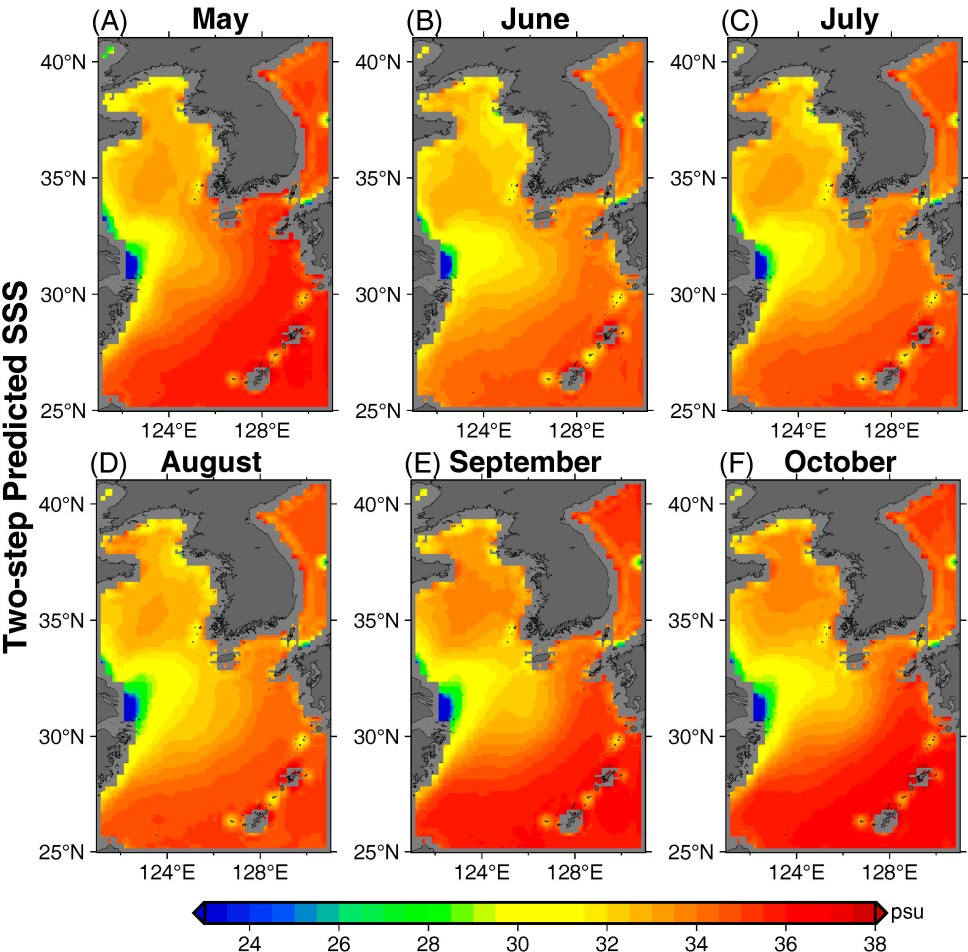

**Figure 11.** Predicted SSSs over Domain 2 for the two-step prediction experiment from May to October.

In conclusion, our U-net model did have the potential to predict further into the future with stable and reliable performance even though the current settings were designed for one-step prediction. Moreover, the above results also suggest that our model predicts the future SSSs mainly based on the temporal evolution of the input SSSs instead of the pre-learned seasonal cycles, which are similar to the regular numerical models that depend on the initial input data.

Note that some previous studies have suggested that neural networks are able to directly perform multi-step prediction (e.g., [68]), rather than the current procedures based on the replacement of the input data. However, with limited data, it is hard to evaluate such experiments in this study, and it also remains unclear which setting is better. More specific experiments with larger datasets are required, which is beyond our current scope and will be the focus of one of our future works.

## 5. Conclusions

In this paper, the deep learning neural network model for predicting (SSS was constructed, based on the U-net method, for the WP region. Our model was designed to use the SSSs of six consecutive months to predict the SSS one month later. The SSSs were obtained from the observational monthly SMAP L3 dataset, while the data from 2015 to 2020 were used for model training and the data for 2021 were used to evaluate the predictability. In general, the predicted SSSs generally reproduced the large-scale distributions of the SSSs over the WP region (i.e., Domain 1). The MEs, RMSEs, MAPEs, and PCCs were about −0.125 psu, 0.7 psu, 1%, and 0.69, respectively. However, most small-scale features were smoothed out in our Domain 1 model, and the model also failed in predicting the spatial and temporal variations over the ECS&YS region, such as the surface freshening and

re-salting processes from summer to autumn. To overcome such problems, we constructed a more specific U-net model for the small ECS&YS region (Domain 2), which successfully predicted both the spatial and temporal variations in SSSs in the ECS&YS region. The RMSEs and MAPEs were reduced by about 30% in the Domain 2 model, and the PCC was also increased to 0.71.

By evaluating the error sources, our results showed that the spatial distribution of SSSs could induce large prediction errors, especially in the regions with extreme values, such as the low SSSs in the ECS&YS region. However, this part of the errors could be reduced through the removal of the long-term mean SSSs in the training data. In addition, our results also suggested that a specific domain could help in predicting the small-scale features, which may be more important than the complexity (hence, the structure) of the U-net model. On the other hand, by evaluating the model performance in multi-step predictions, we found the current U-net model does have the potential for long-time prediction—although the performance for the following months strongly depended on the prediction errors of the first month, which was similar to the ordinary numerical models. Moreover, the above results also show that our model could be further improved when using datasets with higher temporal and spatial resolutions.

Overall, this study introduced a useful neural network, U-net, for predicting the SSSs based on the relatively small amount of trainable data, and we also provided a clear reference and guidance for reducing errors in applications using the U-net and/or other similar deep learning models. Future works will focus on the predictions of other oceanic variables, their three-dimensional structures, and the optimization of training efficiency and accuracy. The combinations of parameters with physical meanings will also be considered in our future works.

**Supplementary Materials:** The following supporting information can be downloaded at: https://www.mdpi.com/article/10.3390/rs15061684/s1.

**Author Contributions:** Conceptualization, X.Z., N.Z. and Z.H.; methodology, X.Z. and Z.H.; software, X.Z.; validation, X.Z.; writing—review and editing, X.Z. and N.Z.; funding acquisition, Z.H. All authors have read and agreed to the published version of the manuscript.

**Funding:** This research was funded by the Scientific Research Program of the Shanghai Science and Technology Commission (grant number 18DZ2253900), the open fund of the Key Laboratory for Information Science of Electromagnetic Waves, Fudan University (grant number EMW201909), and the University–Industry Collaborative Education Program Initiated by the Ministry of Education (grant number 202102245031).

**Data Availability Statement:** SMAP L3 salinity data products are available at www.remss.com (accessed on 10 April 2022).

**Conflicts of Interest:** The authors declare that the research was conducted in the absence of any commercial or financial relationships that could be construed as a potential conflict of interest.

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
