# Peer review of "A Modified U-Net Model for Predicting the Sea Surface Salinity over the Western Pacific Ocean"

_remotesensing, doi:10.3390/rs15061684_

Round 1
Reviewer 1 Report
I was asked to review “A modified U-net model for predicting the sea surface salinity over the western Pacific Ocean” by Zhang et al. which employed a U-net model for the prediction of sea surface salinity (SSS) by training the model on satellite SSS data from consecutive months. The goal of the article was to evaluate the potential ability of the U-net model to represent the spatiotemporal variations of SSS predictions with limited training data. This article showed the advantages of the model's use for SSS prediction but should caution readers of the disadvantages more accurately (i.e. not intended for which purposes or where future studies could improve this type of model).
General:
- The error estimations for predicted SSS in 2021 are concerning. They bring caution towards using this model in the regions suggested.
- The authors refer to the fact that the U-net model has the ability to predict oceanic features throughout the article, but these are mainly concerned with SSS, not any other parameter. I would suggest being more precise and definitive on the wording as the reader may assume that other parameters were involved with this analysis (sea surface temperature, circulation processes, current velocities, etc.)
- The use of the phrase ‘oceanic features’ is a bit broad and encompassing of items more than SSS, I would suggest the authors use a more definitive phrase.
Specific:
Lines 12-13: This is confusing, the authors used six months to train the model, then predicted the next but used 5 years of satellite data to predict the next year?
Line 15: Specify WP (“western Pacific”) instead of having an acronym.
Abstract: The authors suggest the model could be used for long-time prediction and should specify the time length for which this was tested (up to 5 months).
Line 24: Are there other oceanic features that this model was able to predict, other than SSS?
Line 33: Should be a comma after “variabilities” and include “the Earth’s” with a capital E.
Line 51: The period after “al” in “Shi et al”
Line 60: Are there other disadvantages to deep-learning models (other than the sample size of data for training)?
Line 65: Change “in predicting” to “to predict”
Line 68: add ‘of’ before ‘which’
Line 71-75: This could be separated into two sentences for better readability
Lines 80-84: Please add references to back up this statement. Particularly associated with the list describing the importance of why this study region was chosen.
Line 95: “National Aeronautics and Space Administration (NASA)”
Line 101: The authors state “the data after January 2021”, which includes the first few months of 2022. The Abstract states only 2021 data was used for testing. This could be better stated for clarity.
Line 116: This could be a viewing error, but “the convolution” seems to be in different font and size. Please adjust accordingly.
Figure 2: It would be useful to mention that this is satellite-based SSS if this is the case.
Line 163: Wouldn’t omitting raw SSS
Line 169-170: Check font size and font type.
Section 2.4: If the model is inclined to ignore regions of lower salinity and stronger variations would it be safe to assume that this model should not be employed for coastal regions where a river, or low salinity, sources influence the coastal dynamics? This could include prominent areas such as the Andaman Sea dominated by the seasonal monsoons, the Arctic Ocean which accounts for ~11% of the global river discharge, etc.
Lines 226-227: What is it meant by ‘predicting oceanic features’ this is a vague comment and could include the general pattern and variations that the U-net model had difficulty capturing.
Figure 3. I would like to see more reasoning behind the lower salinity values in the predicted SSS (lower panels). What causes this to occur in the U-net model and what suggestions do the authors propose in a solution to this?
Figure 4: Subplot (b) is a bit concerning, the RMSE shows values greater than 0.5. A general rule is that between 0.2 and 0.5 a model is able to predict the data accurately.
Figure 5: As line 252 suggests that the RMSE and MAPE were largely reduced, the MAPE does show a great improvement for highly accurate forecasting. However, the RMSE does not show values in the acceptable range for accuracy.
Figure 6: The mean error of salinity in the open ocean seems acceptable; however, much of the marginal sea regions are equal to or exceed 2 psu which denotes a relatively large error. How do the authors suggest the U-net model be used in situations such as this?
Line 290: “…and how large could it be” if ‘it’ refers to ‘such influences’ it should be plural.
Line 342: SST is not mentioned in other sections of the paper, replace it with “sea surface temperatures”.
Line 370: What do the authors mean by ‘coarser’? This is important to define in more detail for application purposes.
Conclusions: Acronyms do not need to be specified again, ‘SSS’ and ‘WP’ are okay to use alone, just as ‘MEs, RMSEs, MAPE, ECS&YS, etc’ are not again specified.
Reviewer 2 Report
The manuscript introduced U-net in predicting the monthly sea surface salinity for the western Pacific region and the ECS&YS region. And the SSSs obtained from the observational monthly SMAP L3 dataset were used for model training and evaluating. In order to improve the writing and understanding of the article, I request your attention to the following questions:
1. The model uses the SSSs of six consecutive months to predict the SSS one month later. Since the data from 2015 to 2020 were used to calculate the SSS anomaly, why not try some other combinations for model training?
2. The manuscript lacks comparison with other deep learning models, which leads to a not very convincing conclusion.
3. Line 81: Where is Figure 1a?
4. Please check Equation (7) and Equation (9). There is no ‘mean’ at all.
5. Figure 5(A): Why are the true SSSs of Domain 2 missing?
6. Many line charts and the table in Supplementary Material have no units.
Reviewer 3 Report
The manuscript entitled “A modified U-net model for predicting the sea surface salinity over the western Pacific Ocean” discusses the performance of a deep learning model called U-net for predicting sea surface salinity (SSS) in the western Pacific region. The authors examine two different pre-processing methods for the input data, one using anomalies from the long-term mean and the other using the original SSS data, and compare the performance of the model with varying numbers of filters in the first layer.
The authors find that the pre-processing method using SSS anomalies leads to better predictions of SSS than using the original SSS data, particularly over the NECC region and the Kuroshio and its extension region. However, the authors note that this pre-processing method may worsen the prediction in regions dominated by the Central Density Current (CDWC). The authors also find that increasing the number of filters does not always lead to better performance, particularly in predicting relatively small features in a large domain.
Although, some modifications can be done to improve the quality of the manuscript. A detailed review of the manuscript is described below.
The introduction is lacking the novelty and significance of the study. Which must include in the last paragraph. And I don’t see any Fig 1a in the manuscript cited in Lines No 81-82.
The contents in L85 to L90 are not relevant, which may be removed, instead, the authors can highlight the novelty and the significance of the study.
The methodology is well described, however, its lacks proper discussion regarding the model.
Why this model is chosen for the study, and why not other machine learning algorithms? The author should justify the model selection.
The manuscript lacks scientific discussion with respect to the findings, and how this finding is significant.?
Overall, the manuscript provides insights into the performance of deep learning models for predicting SSS in the western Pacific region and highlights the importance of choosing appropriate pre-processing methods and filter numbers for different regions and features.
Round 2
Reviewer 2 Report
The manuscript has been carefully revised,and it can be accepted for publication.